# mTORC1 in the orbitofrontal cortex promotes habitual alcohol seeking

## Nadege Morisot[†], Khanhky Phamluong, Yann Ehinger, Anthony L Berger, Jeffrey J Moffat, Dorit Ron*

Department of Neurology, University of California, San Francisco, San Francisco, United States

**Abstract** The mechanistic target of rapamycin complex 1 (mTORC1) plays an important role in dendritic translation and in learning and memory. We previously showed that heavy alcohol use activates mTORC1 in the orbitofrontal cortex (OFC) of rodents (Laguesse et al., 2017a). Here, we set out to determine the consequences of alcohol-dependent mTORC1 activation in the OFC. We found that inhibition of mTORC1 activity in the OFC attenuates alcohol seeking and restores sensitivity to outcome devaluation in rats that habitually seek alcohol. In contrast, habitual responding for sucrose was unaltered by mTORC1 inhibition, suggesting that mTORC1's role in habitual behavior is specific to alcohol. We further show that inhibition of GluN2B in the OFC attenuates alcohol-dependent mTORC1 activation, alcohol seeking and habitual responding for alcohol. Together, these data suggest that the GluN2B/mTORC1 axis in the OFC drives alcohol seeking and habit.

*For correspondence:
dorit.ron@ucsf.edu

Present address: [†]Charles River Laboratories, South San Francisco, United States

Competing interests: The authors declare that no competing interests exist.

## Introduction

mTORC1 is a multiprotein complex that contains the serine/threonine protein kinase mTOR, the regulatory associated protein of TOR (Raptor), and other enzymes and adaptor proteins (*Lipton and Sahin, 2014*). Upon activation, mTORC1 phosphorylates eIF4E-binding protein (4E-BP) and the ribosomal protein S6 kinase (S6K), which in turn phosphorylates its substrate, S6 (3). These phosphorylation events lead to the translation of a subset of mRNAs to proteins (*Lipton and Sahin, 2014*; *Saxton and Sabatini, 2017*). In the CNS, mTORC1 is responsible for the local dendritic translation of synaptic proteins (*Buffington et al., 2014*; *Santini et al., 2014*). As such, mTORC1 plays an important role in synaptic plasticity, and learning and memory (*Hoeffer and Klann, 2010*).

Increasing lines of evidence in rodents suggest that mTORC1 is a key player in mechanisms underlying alcohol use disorder (AUD) (*Neasta et al., 2014*). For instance, excessive alcohol intake and reinstatement of alcohol place preference activate mTORC1 in the rodent nucleus accumbens (NAc) (*Laguesse et al., 2017a*; *Neasta et al., 2010*; *Beckley et al., 2016*; *Ben Hamida et al., 2019*), resulting in the translation of synaptic proteins, which in turn produce synaptic and structural adaptations that drive and maintain heavy alcohol use and relapse (*Laguesse et al., 2017a*; *Neasta et al., 2010*; *Beckley et al., 2016*; *Ben Hamida et al., 2019*; *Liu et al., 2017*). Repeated cycles of voluntary binge drinking of alcohol and withdrawal also produce a robust and sustained activation of mTORC1 in the OFC of rodents (*Laguesse et al., 2017a*); however, the behavioral consequences of this activation are unknown.

The OFC integrates sensory and reward information (*Wallis, 2007*) and is an essential player in decision making (*Wallis, 2007*; *Meyer and Bucci, 2016*; *Baltz et al., 2018*), in stimulus-outcome association (*Gremel and Costa, 2013*; *Ostlund and Balleine, 2007*; *Gourley et al., 2013*), in updating the value of predicted outcomes (*Baltz et al., 2018*; *Fiuzat et al., 2017*), in goal-directed (*Gremel and Costa, 2013*; *Bradfield et al., 2015*; *Gremel et al., 2016*), and compulsive behaviors (*Ahmari et al., 2013*; *Pascoli et al., 2018*). Exposure to drugs of abuse impairs the performance of

OFC-dependent behavioral tasks (*Schoenbaum and Shaham, 2008*), and aberrant neuroadaptations induced by drugs of abuse in the OFC contribute to drug-seeking and compulsive drug taking (*Schoenbaum and Shaham, 2008*; *Everitt et al., 2007*; *Lüscher, 2016*). For example, withdrawal from alcohol vapor exposure produces morphological and cellular alterations in rodents (*McGuier et al., 2015*; *Nimitvilai et al., 2016*). Inactivation of the OFC enhances alcohol consumption in alcohol-dependent mice (*den Hartog et al., 2016*), and chronic alcohol use alters the protein landscape in the primate OFC (*Nimitvilai et al., 2017*). In humans, degraded OFC white matter and reduced neuronal density are associated with alcohol-dependence (*Pfefferbaum and Sullivan, 2005*; *Miguel-Hidalgo et al., 2006*), and presentation of alcohol-associated cues to alcohol-dependent subjects elicits OFC activation (*Dom et al., 2005*; *Claus et al., 2011*; *Filbey et al., 2008*).

Here, using a rat model system, we set out to examine whether mTORC1 in the OFC contributes to the mechanisms underlying AUD.

## Results

### mTORC1 in the OFC contributes to alcohol seeking

We first examined whether activation of mTORC1 in the OFC contributes to the maintenance of heavy alcohol use. Rats that underwent 7 weeks of intermittent access to 20% alcohol in a 2-bottle choice paradigm (IA-20%2BC) received an intra-OFC bilateral infusion of vehicle or the mTORC1 inhibitor, rapamycin (50 ng/µl), 3 hr prior to a drinking session and alcohol consumption was monitored (*Neasta et al., 2010*) (*Figure 1—figure supplement 1*). As shown in *Figure 1—figure supplement 2* (Source data Figure 1), intra-OFC administration of rapamycin did not impact voluntary intake of alcohol. Next, rats that underwent IA-20%2BC for 7 weeks were trained to self-administer 20% alcohol in an operant self-administration paradigm (*Carnicella et al., 2008*). Upon establishing a baseline of alcohol lever presses, rats received a bilateral infusion of vehicle or rapamycin (50 ng/µl) into the OFC 3 hr prior to an OSA session, and the number of lever presses and alcohol consumption were measured (*Neasta et al., 2010*). As shown in *Figure 1a–c*, Source data Figure 1, inhibition of mTORC1 in the OFC failed to alter the number of lever presses (*Figure 1a–b*, Source data Figure 1) and alcohol consumed (*Figure 1c*, Source data Figure 1). Inactive lever pressing and latencies to the first and last lever presses were also unaltered (*Figure 1—figure supplements 3*, Source data Figure 1). We then determined whether mTORC1 plays a role in alcohol seeking by testing if rapamycin alters lever presses during an extinction session. Intra-OFC administration of rapamycin suppressed the number (*Figure 1d–e*, Source data Figure 1) of rats' responding on the lever that was previously associated with alcohol. The reduction in alcohol seeking was not due to alterations in locomotion, as inter-response intervals were similar in vehicle and rapamycin-treated rats (*Figure 1—figure supplements 4*, Source data Figure 1). Together, these data suggest that mTORC1 in the OFC does not contribute to alcohol drinking per se, but does promote alcohol seeking.

### mTORC1 in the OFC participates in habitual responding for alcohol

Reward seeking can be driven by goal-directed or habitual compulsive behaviors (*Corbit and Janak, 2016*; *Seif et al., 2013*; *Pascoli et al., 2015*). We next examined whether mTORC1 in the OFC plays a role in goal-directed or habitual alcohol seeking using a satiety outcome devaluation procedure. Rats that first underwent 7 weeks of IA-20%2BC were then trained to lever press for 20% alcohol using either a random ratio (RR) or random interval (RI) schedule of reinforcement, which biases lever responding toward goal-directed (RR) or habitual (RI) actions (*Gremel and Costa, 2013*; *Gremel et al., 2016*; *Corbit et al., 2012*) (Timeline. *Figure 2a*). RR and RI trained rats showed a similar number of presses throughout the training (*Figure 2—figure supplements 1a*,Source data Figure 2), however alcohol consumption was slightly higher during RI training (*Figure 2—figure supplements 1b*,Source data Figure 2), presumably due to fewer lever presses being required to obtain the same volume of alcohol, compared with RR training. Rats trained to self-administer alcohol in a goal-directed manner are sensitive to outcome devaluation, while habitually-trained rats are not, i.e. animals who habitually press a lever to obtain a reward will continue to press that lever, even when the relative value of the reward is reduced due to satiety (*Gremel and Costa, 2013*; *Gremel et al., 2016*; *Corbit et al., 2012*; *Rossi and Yin, 2012*). Thus, we conducted a devaluation procedure to differentiate between goal-directed and habitual responding (*Figure 2b*). On the

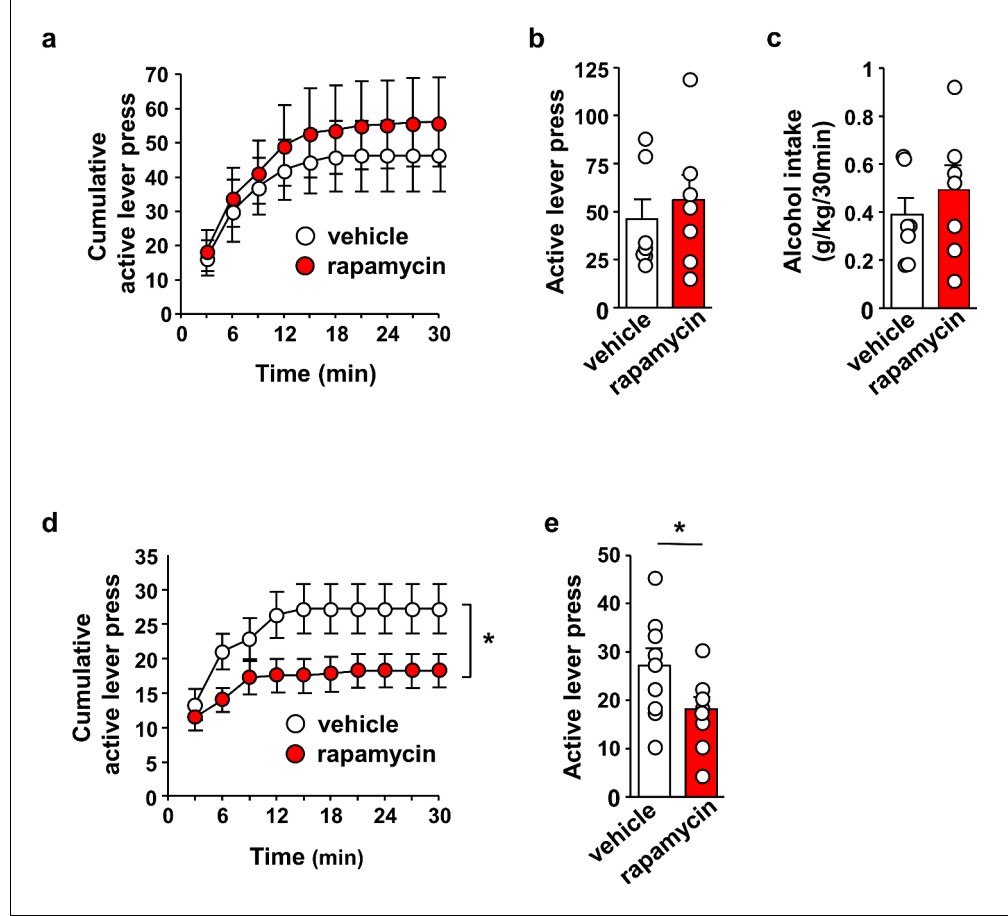

**Figure 1.** Inhibition of mTORC1 in the OFC reduces alcohol seeking. (a–c) Intra-OFC infusion of rapamycin does not alter self-administration of alcohol. Rats underwent 7 weeks of IA-20%2BC and were then trained to self-administer 20% alcohol using a FR3 schedule. Vehicle (white) or rapamycin (50 ng/µl, red) was infused bilaterally in the OFC 3 hr before a 30 min self-administration session. (a) RM ANOVA of cumulative lever presses did not identify a significant main treatment effect ($F_{1,12}=0.26$, $p>0.05$). Two-tailed paired t-test revealed that the number of active lever presses (b) ($t_6 = 1.40$, $p>0.05$), and amount of alcohol consumed (c) ($t_6 = 1.51$, $p>0.05$) did not differ between treatment groups. (d–e) Intra-OFC infusion of rapamycin inhibits lever presses during extinction. Vehicle (white) or rapamycin (50 ng/µl, red) was infused in the OFC 3 hr before a 30 min extinction session, and responses on the previously active lever were recorded. (d) RM ANOVA revealed a significant main treatment effect ($F_{1,16}=4.40$, $p=0.05$) of cumulative lever presses. (e) Two-tailed paired t-test revealed that the number of lever presses (per 5 min) $t_8 = 3.31$, $p<0.05$, were reduced in the rapamycin-treated animals. Data are presented as individual values and mean ± SEM. *$p<0.05$. (a–c) n = 7, (d–e) n = 9.

The online version of this article includes the following source data and figure supplement(s) for figure 1:

**Source data 1.** Cumulative lever presses at 3 min intervals for vehicle- and rapamycin-treated rats during a 30 min self-administration session (*Figure 1a*).

**Source data 2.** Total lever presses and alcohol consumed (g/kg) during a 30 min self-administration session in vehicle- and rapamycin-treated rats (*Figure 1b–c*).

**Source data 3.** Cumulative lever presses at 3 min intervals for vehicle- and rapamycin-treated rats during a 30 min extinction session (*Figure 1d*).

**Source data 4.** Total lever presses during a 30 min extinction session in vehicle- and rapamycin-treated rats (*Figure 1e*).

**Figure supplement 1.** Schematic drawing of cannulae placement.

**Figure supplement 2.** Inhibition of mTORC1 in the OFC does not alter alcohol consumption in a 2-bottle choice paradigm.

**Figure supplement 2—source data 1.** Alcohol consumed ( g/kg) by vehicle- and rapamycin-treated rats after 30 min or 24 hr of two-bottle choice.

*Figure 1 continued on next page*

*Figure 1 continued*

**Figure supplement 3.** Inhibition of mTORC1 in the OFC does not impact inactive lever presses or the latency to the first and last active lever press.
**Figure supplement 3—source data 1.** Total inactive lever presses and the latencies to the first and last active lever presses during a 30 min self-administration session in vehicle- and rapamycin-treated rats.
**Figure supplement 4.** Intra-OFC infusion of rapamycin does not alter locomotion.
**Figure supplement 4—source data 1.** Inter-response intervals during a 30 min self-administration session in vehicle- and rapamycin-treated rats.

devalued (D) day, rats had 30 min of home-cage access to 20% alcohol, the reward previously associated with lever responding, and on the non-devalued (ND) day, rats had 30 min of home-cage access to 1% sucrose, a reward that was not previously associated with lever responding (*Figure 2b*). Rats then underwent a 10 min extinction test, during which lever presses were recorded (*Figure 2b*). In order to minimize the effects of individual variability in overall lever pressing, and to graphically represent the distribution of lever presses between ND and D, we normalized lever-pressing data such that 50% indicates identical lever pressing on both days as described in *Baltz et al. (2018)*; *Gremel and Costa (2013)*; *Gremel et al. (2016)*. As shown in *Figure 2c* (Source data Figure 2), rats trained on a RR schedule exhibit a decrease in responding following alcohol devaluation, while RI-trained rats remained insensitive to devaluation. These data show that RR training produces goal directed alcohol seeking whereas RI training produces habitual responding for alcohol.

Next, to test whether mTORC1 in the OFC participates in goal-directed or habitual alcohol seeking, rapamycin (50 ng/μl) or vehicle were infused bilaterally, 3 hr prior to a devaluation session, and lever presses in RR- and RI-trained groups were examined (*Figure 2a*, *Figure 2—figure supplement 2*, *Table 1*, Source data Figure 2). Rapamycin administration did not alter home cage alcohol intake prior to the extinction session in either group (*Figure 2—figure supplements 3—source data 1*, *Figure 2*). Intra-OFC administration of rapamycin did not impact responding in RR-trained (goal-directed) cohorts, as lever presses were reduced in rats that were pre-fed with alcohol and treated with either vehicle or rapamycin (*Figure 2d*, Source data Figure 2). In contrast, inhibition of mTORC1 in the OFC reduced responding in the RI-trained (habitual) rats in the devalued condition (*Figure 2e*, Source data Figure 2). Critically, the reduction in responding in the rapamycin-treated RI-trained rats paralleled the responding of RR-trained animals (*Figure 2d vs. 2e*, Source data Figure 2). These data suggest that mTORC1 inhibition in the OFC restores goal-directed alcohol seeking in animals trained to habitually self-administer alcohol.

To determine whether mTORC1 in the OFC plays a role in habitual responding to a natural reward, animals were trained to self-administer sucrose under an RI schedule of reinforcement, and lever presses were measured in rats treated with rapamycin (50 ng/μl) or vehicle (*Figure 3—figure supplement 1*) during an extinction session in animals that were first offered sucrose (devalued) or alcohol (non-devalued) in their home cage (Timeline, *Figure 3a–b*). Rapamycin administration did not alter home cage sucrose intake prior to the extinction session (*Figure 3—figure supplement 2*, *Table 1*, Source data Figure 3). As shown in *Figure 3c* (Source data Figure 3), habitual responding for sucrose was similar in the vehicle- and rapamycin-treated animals. Together, these data suggest that mTORC1's contribution to habitual behavior is not generalized to all rewarding substances.

## Alcohol-dependent mTORC1 activation in the OFC requires GluN2B

Next, we set out to elucidate the mechanism by which long-term heavy alcohol use activates mTORC1 in the OFC. Glutamatergic inputs from cortical and subcortical structures project to the OFC (*Schoenbaum et al., 2003*; *Lichtenberg et al., 2017*). Extracellular glutamate content is elevated in cortical regions of humans and rats during alcohol withdrawal (*Hermann et al., 2012*). Glutamate binds the N-methyl D-Aspartate receptor (NMDAR), which is a known target of alcohol (*Morisot and Ron, 2017*). For example, alcohol exposure promotes the phosphorylation of the GluN2B subunit of the NMDA receptor in the hippocampus and dorsomedial striatum (DMS) resulting in the activation of GluN2B-containing NMDARs in these two brain regions (*Morisot and Ron, 2017*). Stimulation of GluN2B activates the guanine nucleotide releasing factor 1 (GRF1), the activator of H-Ras (*Krapivinsky et al., 2003*) (*Figure 4—figure supplement 1*). Alcohol activates H-Ras in

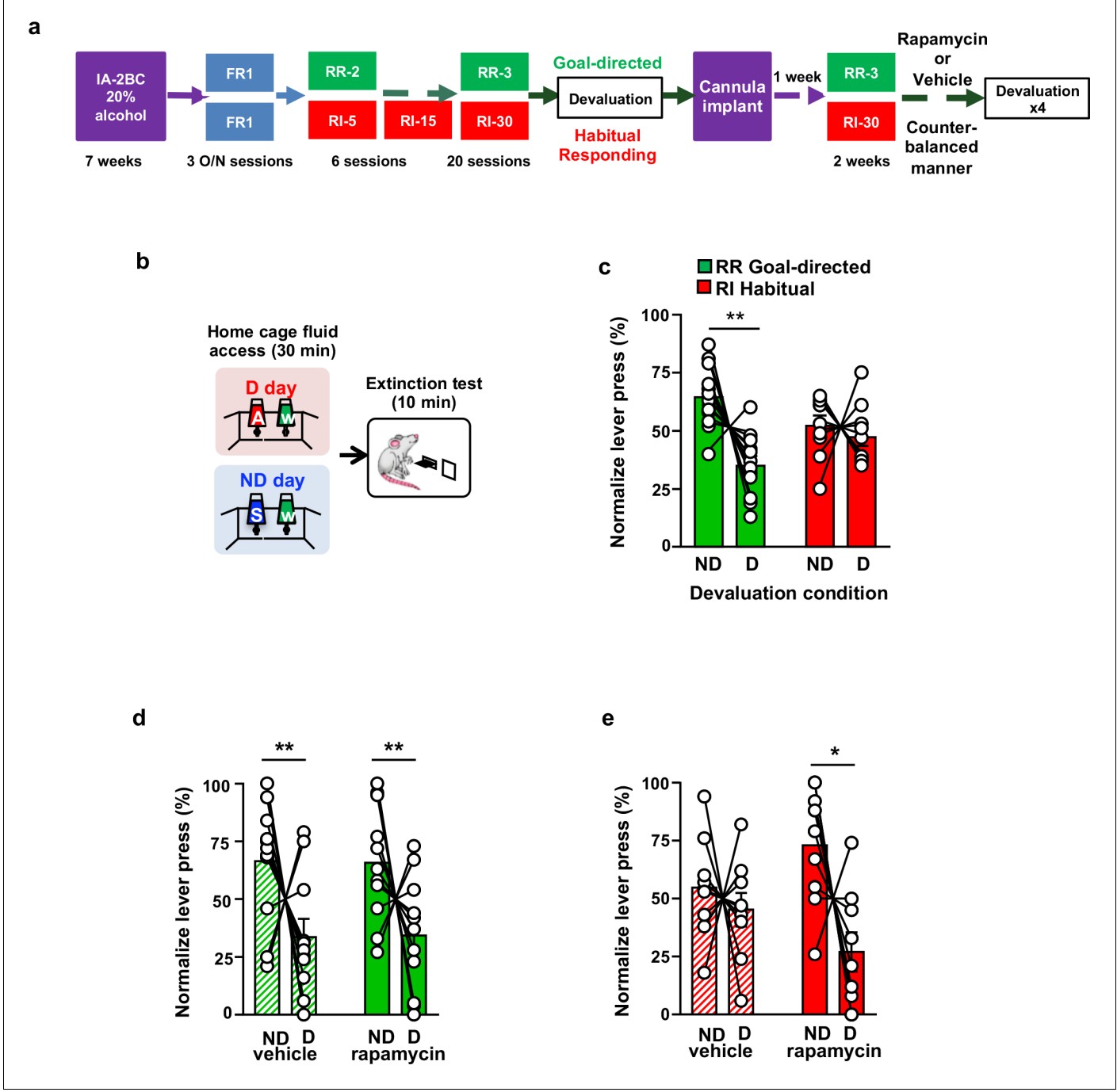

**Figure 2.** Inhibition of mTORC1 in the OFC attenuates habitual responding for alcohol. (a) Timeline of experiment. Rats underwent 7 weeks of IA20%—2BC. Rats were then trained to operant self-administer 20% alcohol during three overnight FR1 sessions and were pseudo-randomly assigned to two groups. One group was subjected to a progressive RI reinforcement schedule, while the other was assigned to a RR schedule. Following a stable RI-30 or RR-3 responding, rats underwent alcohol devaluation testing depicted in (b). Upon establishing the behavior, rats underwent stereotaxic surgery to bilaterally implant a guide cannula. One week later, RI-30 or RR-3 training was resumed for 2 weeks, after which rats received microinjections of vehicle or drug in a counter-balanced manner, prior to the devaluation test. (b) Alcohol devaluation test. On test day, rats were pre-fed with 20% alcohol for 30 min in their home cage during devalued day (D) or with 1% sucrose in the non-devalued day (ND) (*Table 1*), and normalized lever pressing (percent of total presses on D or ND, see Materials and methods) was used to evaluate goal-directed or habitual responding during a 10 min extinction session. (c) RR trained animals are goal-directed whereas RI trained animals show habitual behavior. RR trained rats (green) press less during devalued day as compared to RI trained animals (red). Two-way ANOVA with Sidak post hoc analysis indicates a significant decrease in normalized lever pressing on D vs ND in RR-trained rats (p=0.0018), but not in RI-trained rats (p=0.8014) (Main effect of devaluation: $F_{(1, 20)}$=9.391, p=0.0061; reinforcement schedule X devaluation: $F_{(1, 20)}$=4.736, p=0.0417). (d) Intra-OFC infusion of rapamycin does not alter goal-directed alcohol seeking. Rats that were trained using

*Figure 2 continued on next page*

*Figure 2 continued*

a RR schedule of reinforcement received an intra-OFC infusion of vehicle (hatched green) or rapamycin (50 ng/µl) (green) 3 hr before the alcohol devaluation procedure as in (**b**), and the number of lever presses were measured during extinction. Total lever press-based normalization indicates significant decreases in lever pressing between ND and D days in both vehicle-treated and rapamycin-treated RR-trained rats (Main effect of devaluation: $F_{(1, 20)}$=8.826, p=0.0076). (**e**) Intra-OFC infusion of rapamycin reduces habitual lever presses. Rats that were trained using a RI schedule of reinforcement received an intra-OFC infusion of vehicle (hatched red) or rapamycin (50 ng/µl) (red) 3 hr before the alcohol devaluation procedure as in (**b**), and the number of lever presses were measured during extinction. Total lever press-based normalization indicates significant decreases upon intra-OFC infusion of rapamycin- (p=0.0202), but not vehicle (p=0.8006) in RI trained rats on D compared to ND days (Main effect of devaluation: $F_{(1,16)}$ =6.189, p=0.0243). Data are presented as mean ± SEM. *p<0.05, **p<0.01 . (**c**) n = 8–10, (**d**) n = 11, (**e**) n = 9.

The online version of this article includes the following source data and figure supplement(s) for figure 2:

**Source data 1.** Total lever presses during 10 min extinction sessions on non-devalued and devalued days in RR- and RI-trained rats (*Figure 2c*).

**Source data 2.** Total lever presses during 10 min extinction sessions on non-devalued and devalued days in vehicle- and rapamycin-treated, RR-trained rats (*Figure 2d*).

**Source data 3.** Total lever presses during 10 min extinction sessions on non-devalued and devalued days in vehicle- and rapamycin-treated, RI-trained rats (*Figure 2e*).

**Figure supplement 1.** Number of lever presses and alcohol consumed in RI and RR-trained animals.

**Figure supplement 1—source data 1.** Total lever presses during 1 hr RR and RI training sessions (*Figure 2—figure supplements 1a*).

**Figure supplement 2.** Schematic drawing of cannulae placement.

**Figure supplement 3.** Inhibition of mTORC1 in the OFC does not alter voluntary alcohol intake prior to devaluation.

**Figure supplement 3—source data 1.** Alcohol consumed (g/kg) during 30 min home cage alcohol exposure prior to devaluation extinction tests in vehicle- and rapamycin-treated rats.

the NAc (*Ben Hamida et al., 2012*), as well as AKT in striatal regions (*Neasta et al., 2011*; *Laguesse et al., 2018*), and in the OFC (*Laguesse et al., 2017a*). Since the H-Ras/AKT axis is upstream of mTORC1 (*Manning and Toker, 2017*) (*Figure 4—figure supplement 1*), we hypothesized that alcohol stimulates GluN2B in the OFC resulting in mTORC1 activation.

First, we examined whether alcohol increases GluN2B phosphorylation in the OFC. Rats were subjected to 7 weeks of IA-20%2BC and the phosphorylation of GluN2B was measured at the end of last 24 hr of deprivation session (*Table 2*). Rats that drank water only for the same duration of time were used as control. As shown in *Figure 4a–b*, Source data Figure 4a-b, alcohol drinking and deprivation produced a robust increase in GluN2B phosphorylation. In contrast, the total level of GluN2B was unaltered by alcohol exposure (*Figure 4a–b*, Source data Figure 4a-b). Together, these data suggest that repeated cycles of alcohol binge and withdrawal enhances the activity of GluN2B containing NMDARs in the OFC.

Next, we tested whether NMDAR stimulation activates mTORC1 in the OFC. To do so, OFC slices were briefly treated with NMDA (25 µM), and the activation of CaMKII and mTORC1 was tested by measuring the phosphorylation of CaMKII and of mTORC1 downstream targets, S6K and S6 (*Figure 4—figure supplement 1*). We found that ex vivo treatment of the OFC with NMDA increased the phosphorylation, and thus activation, of CaMKII, a marker of NMDAR activation (*Figure 4c–d*, Source data Figure 4c-d). In parallel the phosphorylation of S6K and S6 were also increased (*Figure 4c–d*, Source data Figure 4c-d). These data suggest that NMDAR stimulation activates mTORC1 in the OFC.

We previously showed that mTORC1 is activated in the OFC in response to a binge drinking session which persists for at least 24 hr into alcohol abstinence (*Laguesse et al., 2017a*). Thus, we next tested whether GluN2B is required for alcohol-dependent mTORC1 activation during alcohol deprivation. To do so, rats that underwent 7 weeks of IA-20%2BC paradigm or that drank water only for the same duration of time, received a bilateral intra-OFC infusion of the selective GluN2B antagonist Ro-25–6981 (*Fischer et al., 1997*) (10 µg/µl) or vehicle, 15 min prior to the end of the last alcohol drinking session. mTORC1 activation was examined at the end of the last 24 hr alcohol deprivation session by measuring the phosphorylation of mTORC1 downstream targets, 4EBP and S6 (*Figure 4—figure supplement 1*, Timeleine *Figure 4—figure supplement 2*, *Table 2*). In vehicle-treated rats, alcohol produced a robust activation of CaMKII (*Figure 4e,h*, Source data Figure 4e-j), and mTORC1, as measured by S6 (*Figure 4f,I*, Source data Figure 4e-j), and 4E-BP phosphorylation (*Figure 4g,j*, Source data Figure 4e-j). In contrast, R025-6981-treated rats exhibited a marked reduction in alcohol-dependent CaMKII, S6 and 4E-BP phosphorylation (*Figure 4e–j*, Source data Figure

**Table 1.** Mean consumption of sucrose, alcohol and water during outcome devaluation.
Rats undergoing alcohol seeking probe trials were pre-fed with sucrose (non-devalued test) or alcohol (devalued test), and water 30 min prior to a 10 min extinction session.

**Alcohol Devaluation**

|  | Training | RI | | RR | | RI | |
|---|---|---|---|---|---|---|---|
|  | Group | Vehicle | rapamycin | Vehicle | rapamycin | Vehicle | Ro25 |
| Non-devalued day | 1% Sucrose (ml/kg) | 1.4 | 2.0 | 1.2 | 1.9 | 2.7 | 5.8 |
|  |  | ± | ± | ± | ± | ± | ± |
|  |  | 0.5 | 0.8 | 0.4 | 0.6 | 1.5 | 1.5 |
|  | Water (ml/kg) | 0.7 | 0.6 | 1.2 | 0.8 | 1.0 | 1.3 |
|  |  | ± | ± | ± | ± | ± | ± |
|  |  | 0.3 | 0.1 | 0.7 | 0.1 | 0.2 | 0.3 |
| Devalued day | 20% Alcohol (g/kg) | 0.7 | 0.7 | 1.0 | 0.7 | 1.0 | 1.3 |
|  |  | ± | ± | ± | ± | ± | ± |
|  |  | 0.1 | 0.1 | 0.1 | 0.1 | 0.1 | 0.1 |
|  | Water (ml/kg) | 0.4 | 0.6 | 0.7 | 0.6 | 1.0 | 0.9 |
|  |  | ± | ± | ± | ± | ± | ± |
|  |  | 0.1 | 0.1 | 0.1 | 0.1 | 0.4 | 0.2 |

**Sucrose Devaluation**

|  | Training | RI | |
|---|---|---|---|
|  | Group | Vehicle | rapamycin |
| Non-devalued day | 20% Alcohol (g/kg) | 0.9 | 0.8 |
|  |  | ± | ± |
|  |  | 0.1 | 0.1 |
|  | Water (ml/kg) | 0.9 | 1.1 |
|  |  | ± | ± |
|  |  | 0.1 | 0.1 |
| Devalued day | 1% Sucrose (ml/kg) | 4.5 | 5.1 |
|  |  | ± | ± |
|  |  | 0.7 | 1.4 |
|  | Water (ml/kg) | 1.2 | 0.9 |
|  |  | ± | ± |
|  |  | 0.2 | 0.1 |

4e-j). Together our data suggest that alcohol stimulates GluN2B which in turn promotes mTORC1 activation.

## GluN2B in the OFC is required for alcohol seeking and habit

We reasoned that if GluN2B is recruited in response to alcohol exposure to activate mTORC1, then GluN2B should also participate in habitual alcohol seeking. First, we examined whether inhibition of GluN2B alters alcohol seeking during extinction. We found that intra-OFC administration of Ro25-6981 (10 μg/μl) 15 min prior to an extinction session (*Figure 5—figure supplement 1*) suppressed alcohol seeking, as indicated by a reduction in cumulative (*Figure 5a*, Source data Figure 5), active lever presses (*Figure 5b*, Source data Figure 5), as well as the rate of lever pressing (*Figure 5c*, Source data Figure 5). The reduction of alcohol seeking by Ro25-6981 was not due to alteration of locomotion as inter-response intervals were similar in vehicle-treated and Ro25-6981-treated animals (*Figure 5—figure supplement 2*, Source data Figure 5).

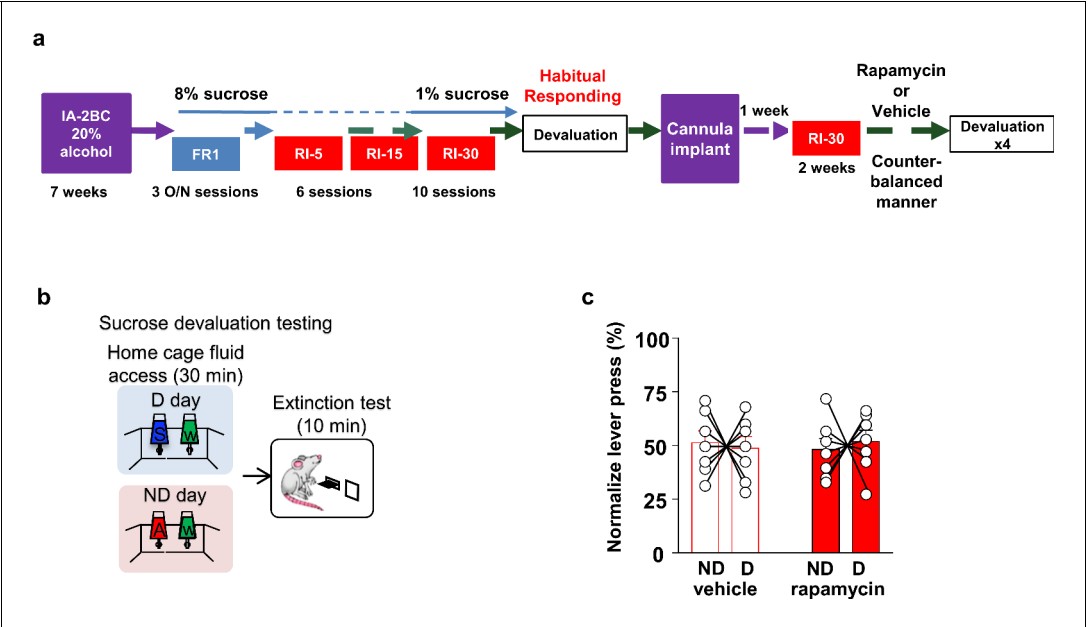

**Figure 3.** Habitual sucrose responding is not affected by mTORC1 inhibition in the OFC. (**a**) Timeline of experiment. Rats underwent 7 weeks of IA20%−2BC and were then trained to operant self-administer 8% sucrose. Sucrose concentration was progressively reduced to 1% during three overnight FR1 sessions. Rats were then subjected to a RI reinforcement schedule. Drinking data are detailed in *Table 1*. Following a stable RI-30 responding, rats underwent alcohol devaluation testing depicted in (**b**). Upon establishing the behavior, rats underwent stereotaxic surgery to bilaterally implant a guide cannula. One week later, RI-30 training was resumed for 2 weeks, after which rats received microinjections of vehicle or rapamycin (50 ng/µl) in a counter-balanced manner, prior to the devaluation test. (**b**) Sucrose devaluation test. Habitual behavior was probed in an identical way to the alcohol-trained cohorts with the exception that sucrose pre-feeding was considered the devaluing context and alcohol was the non-devalued substance. (**c**) Two-way RM ANOVA failed to show a significant effect of repeated testing (Main effect of devaluation: $F_{(1, 12)}=0.006464$, $p>0.9999$; treatment X devaluation: $F_{(1, 12)}=0.7257$, $p=0.4110$) on lever presses between devalued and non-devalued days when habitually responding animals were pretreated with either vehicle or rapamycin. n = 7.

The online version of this article includes the following source data and figure supplement(s) for figure 3:

**Source data 1.** Lever presses during sucrose devaluation extinction test in vehicle- and rapamycin-treated rats (*Figure 3c*).
**Figure supplement 1.** Schematic drawing of cannulae placement.
**Figure supplement 2.** Inhibition of mTORC1 in the OFC does not alter voluntary sucrose intake prior to devaluation.

Finally, we tested whether GluN2B in the OFC contributes to habitual alcohol seeking. The timeline of the experiment is depicted in *Figure 5d*. Intra-OFC infusion of R025-6981 (5 µg/µl) did not impact home cage alcohol consumption during the devalued days (*Figure 5—figure supplement 3*, *Table 1*, Source data Figure 5). In contrast, administration of Ro25-6981 (5 µg/µl) into the OFC shifts habitually responding rats to be more goal-directed in their responding for alcohol (*Figure 5e*, Source data Figure 5). Together, these data indicate that the recruitment of GluN2B signaling by alcohol may be involved in driving alcohol seeking and habit.

## Discussion

Using operant training protocols that bias alcohol seeking towards habitual or goal-directed behavior, we present data implicating mTORC1 in the OFC in mechanisms underlying the development and/or maintenance of habitual behavior associated with alcohol use. In addition, we show that GluN2B in the OFC is required for alcohol-dependent mTORC1 activation and function. Together, our data suggest that the GluN2B/mTORC1 axis acts as a molecular switch that converts the OFC from driving goal directed to habitual alcohol use.

We found that inhibition of mTORC1 in the OFC attenuates lever pressing during extinction, suggesting that mTORC1 regulates alcohol seeking. It is important to note, however, that mTORC1 inhibition does not appear to affect lever pressing in reinforced self-administration sessions. The OFC has been implicated in updating reward anticipation based on the value of an expected outcome

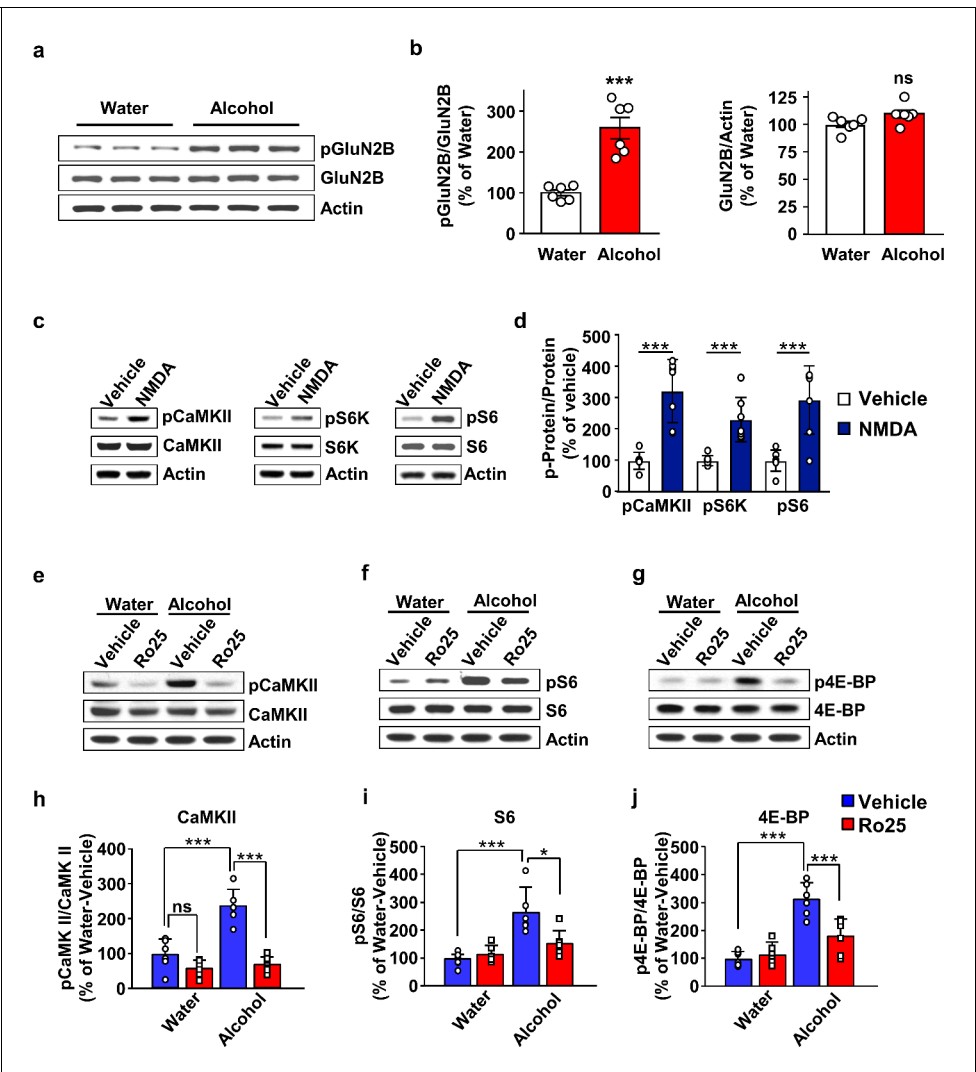

**Figure 4.** GluN2B in the OFC is required for mTORC1 activation by alcohol. (**a–b**) Alcohol increases GluN2B phosphorylation in the OFC. Rats underwent 7 weeks of IA-20%2BC resulting in an average of alcohol intake of 5.01 g/kg/24 hr (**Table 2**). The OFC was dissected at the end of the last 24 hr alcohol withdrawal session, and GluN2B phosphorylation and protein levels were determined by western blot analysis, with actin used as a loading control. (**a**) Representative images depicting GluN2B phosphorylation (top) and total protein (middle) in the OFC of alcohol consuming vs. water only consuming rats. (**b**) ImageJ was used for optical density quantification. Data are expressed as the average ratio ± S.E.M of phospho-GluN2B to GluN2B (middle) or GluN2B to actin (right) and expressed as percentage of the water control. Significance was determined using unpaired t-test. Alcohol increased GluN2B phosphorylation ($t_{10}$ = 5.829, p<0.001), but did not alter GluN2B levels ($t_{10}$ = 2.025, p>0.05). (**c–d**) Stimulation of NMDA receptors activates mTORC1 in the OFC. Vehicle (0.1% DMSO) or NMDA (25 µM) was applied to OFC slices of naïve rats for 3 min. Phosphorylation of CaMKII, S6K and S6 were determined by western blot analysis. Total levels of the proteins were also measured, and actin was used as a loading control. (**c**) Representative blots of CaMKII (left), S6K (middle), and S6 (right) phosphorylation in OFC slices treated with vehicle or NMDA. (**d**) ImageJ was used for optical density quantification. Data are expressed as the average ratio ± S.E.M of phospho-CaMKII to CaMKII, phospho-S6K to S6K and phospho-S6 to S6, and are expressed as percentage of vehicle. Significance was determined using two-tailed unpaired *t*-test. (CaMKII: $t_{12}$ = 5.61, p<0.001; S6K: $t_{12}$ = 4.77, p<0.001; S6: $t_{12}$ = 4.51, p<0.001). (**e–j**) Alcohol-dependent mTORC1 activation is attenuated by the GluN2B inhibitor, Ro25-6981. Rats underwent 7 weeks of IA-20%2BC resulting in an average of alcohol intake of 4.37 g/kg/24 hr (**Table 2**). Rats received an intra-OFC bilateral infusion of vehicle (saline) or Ro25-6981 (Ro25, 10 µg/µl), 15 min prior to the beginning of the last water only session and the OFC was dissected 24 hr later. Water only consuming rats were used as controls. (**e–g**) Representative images of CaMKII (**e**), S6 (**f**) and 4E-BP (**g**) phosphorylation in water vs. alcohol-exposed animals that were pre-treated with vehicle or Ro25-6981. (**h–j**) ImageJ was used for optical density quantification. Data are expressed as the average ratio ± S.E.M of phospho-CaMKII to CaMKII (**h**), phospho-S6 to S6 (**i**), and phospho-4E-BP to 4E-BP (**j**), and are expressed as percentage of water + vehicle group. Two-way RM ANOVA showed a significant effect of alcohol on phospho-CaMKII ($F_{1,20}$=24.66, p<0.0001), phospho-S6 ($F_{1,20}$=21.92, p=0.0001), and phospho-4E-BP ($F_{1,20}$=48.23, p<0.0001). In addition, two-way RM ANOVAs indicated a main effect of drug treatment for phospho-CaMKII ($F_{1,20}$=48.51, p<0.0001), phospho-S6 ($F_{1,20}$=4.58, p<0.05), and phospho-4E-BP ($F_{1,20}$=8.49, p<0.01), as well as treatment x alcohol interaction for phospho-CaMKII ($F_{1,20}$=18.57, p<0.01), phospho-S6 ($F_{1,20}$=7.559, p<0.05), and phospho-4E-BP ($F_{1,20}$=13.22, p<0.01). Tukey's multiple comparison post

*Figure 4 continued on next page*

*Figure 4 continued*

hoc analysis revealed that alcohol increased phospho-CaMKII (p<0.001), pS6 (p<0.001), and phospho-4E-BP (p<0.001) in vehicle-treated animals, and that pretreatment with Ro25-6981 reduced the phosphorylation of CaMKII (p<0.0001), S6 (p<0.05), and 4E-BP (p<0.001). *p<0.05, ***p<0.001. (**a–b, e–j**) n = 6 per group, (**c–d**) n = 7 per group.

The online version of this article includes the following source data and figure supplement(s) for figure 4:

**Source data 1.** Full, uncropped western blot films from experiments in *Figure 4*.
**Figure supplement 1.** GluN2B-dependent activation of mTORC1 signaling.
**Figure supplement 2.** Timeline of experiments depicted in *Figure 4e–j*.

(*Ostlund and Balleine, 2007*; *Schoenbaum et al., 1998*), and O'Doherty et al. showed that the lateral OFC (lOFC) is activated during reward anticipation in humans (*O'Doherty et al., 2002*). Thus, it is plausible that one of the roles of mTORC1 in the OFC is the strengthening of context-alcohol learning, consequently promoting alcohol seeking.

We found that blockade of mTORC1 activity in the OFC converts habitual alcohol responding to a more goal directed behavior. This finding could explain, at least in part, the decrease in alcohol seeking behavior following mTORC1 inhibition. Either extended operant training (*Corbit et al., 2012*) or specific schedules of reinforcement (*Baltz et al., 2018*; *Gremel and Costa, 2013*) have been utilized previously to bias animals toward habitual seeking for alcohol and other substances. By first cultivating habitual alcohol seeking and then testing the effects of rapamycin, we were able to demonstrate that mTORC1 inhibtion alters rat action selection strategies; either enhancing goal-directed responding or inhibiting habitual actions. Habitual behavior is thought to rely on the strengthening of the dorsolateral striatum (DLS) over the DMS, a brain region which participates in goal-directed behaviors (*Corbit et al., 2012*; *Belin et al., 2013*; *Everitt and Robbins, 2005*). Renteria and colleagues showed that alcohol vapor exposure generates habitual behavior via the

**Table 2.** Individual alcohol drinking data from rodents used for biochemical experiments.
Individual alcohol drinking data of the final 4 sessions of IA-2BC20%. Alcohol intake is expressed as mean ± S.E.M.

| *Figure 4a, b* | Drinking paradigm IA20%–2BC 24h-withdrawal | Rat number | Last four drinking session (g/kg /24 ) |
|---|---|---|---|
| | | 1 | 5.97 |
| | Animal numbers | 2 | 6.40 |
| | n = 6 | 3 | 4.00 |
| | | 4 | 5.84 |
| | | 5 | 4.23 |
| | | 6 | 3.60 |
| | | Mean ± S.E.M. | 5.01 ± 0.49 |
| *Figure 4e,f and g* | Drinking paradigm IA20%–2BC 24h-withdrawal | Rat number | Last four drinking session (g/kg/24 h) |
| | Animal numbers n = 12 | 1 | 5.35 |
| | | 2 | 5.52 |
| | | 3 | 5.54 |
| | | 4 | 4.84 |
| | | 5 | 6.41 |
| | | 6 | 3.16 |
| | | 7 | 4.37 |
| | | 8 | 3.45 |
| | | 9 | 3.48 |
| | | 10 | 6.24 |
| | | 11 | 2.71 |
| | | 12 | 5.72 |
| | | Mean ± S.E.M. | 4.73 ± 0.37 |

disruption of OFC to DMS circuit (*Renteria et al., 2018*). Thus, it is plausible that mTORC1 activation is weakening OFC to DMS projections. Furthermore, Gremel et al. showed that the endocannabinoid/CB1 receptor system in OFC to dorsal striatum cicruitry promotes a shift from goal directed behavior to habitual responding (*Gremel et al., 2016*). Interestingly, cannabinoid signaling has been shown to activate mTORC1 in hippocampal neurons (*Puighermanal et al., 2009*). Furthermore, THC-dependent impairment of novel object recognition was reversed by the mTORC1 inhibitor, rapamycin (*Puighermanal et al., 2009*). Thus, it is possible that endocannabinoids and mTORC1 are part of the same signaling cascade that gates goal-directed behaviors, thereby promoting habit. In addition, Gourley and colleagues reported that the brain-derived neurotrophic factor (BDNF) in the OFC participates in goal-directed behaviors (*Gourley et al., 2013*; *Gourley et al., 2016*; *Zimmermann et al., 2017*). BDNF and mTORC1 play opposing roles in AUD (*Ron and Barak, 2016*). For example, BDNF keeps alcohol intake in moderation, and mTORC1 is associated with excessive alcohol use (*Ron and Barak, 2016*). As breakdown of the BDNF signaling cascade results in compulsive alcohol use (*Warnault et al., 2016*). It is tempting to speculate that the balance between BDNF and mTORC1 signaling in the OFC determines whether alcohol use is kept in moderation or becomes habitual.

We previously showed that mTORC1 activation is detected in the lOFC (*Laguesse et al., 2017a*), which was targeted in the current study. However, the structure of the OFC is complex. The medial OFC (mOFC) and lOFC are each part of anatomically and functionally distinct corticostriatal circuits in humans (*Fettes et al., 2017*), and in rodents (*Izquierdo, 2017*). The mOFC has been reported to play a larger role in determining, updating, and storing the relative reward values of stimuli and actions (*Noonan et al., 2010*). The lOFC, on the other hand, is proposed to be more involved in encoding the relationship between stimuli/actions and outcomes (*Noonan et al., 2010*), cue-guided behavioral flexibility (*Panayi and Killcross, 2018*), and goal-directed behavior (*Gremel and Costa, 2013*; *Parkes et al., 2018*). Thus a careful examination of mTORC1's role in lOFC vs. mOFC-dependent behaviors is required.

The OFC has been associated with compulsive behavior (*Ahmari et al., 2013*), and recently Pascoli et al. reported that OFC neurons projecting to the dorsal striatum are associated with compulsive lever pressing in response to optogenetic activation of dopaminergic ventral tegmental area neurons (*Pascoli et al., 2018*). Compulsive drug use is associated with maladaptive habitual responding (*Everitt and Robbins, 2005*; *Smith and Laiks, 2018*). We and others provided animal data to support the notion that alcohol consumption can become compulsive (*Seif et al., 2013*; *Warnault et al., 2016*; *Augier et al., 2018*). Thus, it is possible that mTORC1 signaling in the OFC is driving both habitual and compulsive alcohol seeking.

Interestingly, mTORC1 inhibition in the OFC does not alter habitual responding for sucrose, indicating that mTORC1 plays a specific role in the formation of habits in response to alcohol exposure and not natural rewards. The difference between the consequences of mTORC1 inhibition on habitual alcohol vs. sucrose seeking is striking, but it is important to note that, unlike alcohol, voluntary sucrose intake does not activate mTORC1 in the OFC (*Laguesse et al., 2017a*).

Corbit et al. showed that RR-trained rats exhibit habitual alcohol self-administration after 8 weeks of training on an RR-3 schedule of reinforcement (*Corbit et al., 2012*). In the current study, rats undergoing RR training demonstrated persistent goal-directed alcohol seeking behaviors. The difference between the two studies is most likely to be due to the length of RR training (long in Corbit et al. (*Corbit et al., 2012*) vs. short herein). It is possible that further training would drive RR-trained goal-directed rats to exhibit more habitual phenotypes, and further research would be required to compare the models more accurately.

The NMDAR is a target of alcohol (*Morisot and Ron, 2017*), and alcohol enhances GluN2B phosphorylation in brain regions such as the cerebellum, the DMS, and the hippocampus (*Morisot and Ron, 2017*). GluN2B phosphorylation enhances the activity of the channel (*Trepanier et al., 2012*), and we show that alcohol increases GluN2B phosphorylation in the OFC. Thus, it is plausible that the recruitment of glutamatergic signaling by alcohol in the OFC enhances GluN2B function thereby enabling mTORC1 activation. Nimitvilai and colleagues showed that chronic exposure to alcohol vapor decreases in the expression of GluN2B (*Nimitvilai et al., 2016*), whereas we did not observe any changes in the protein levels of GluN2B following repeated cycles of alcohol binge drinking sessions and withdrawal. One important difference between our study and Nimitvilai et al. as well as additional reports discussed herein, relates to the use of contingent vs. non-contingent alcohol

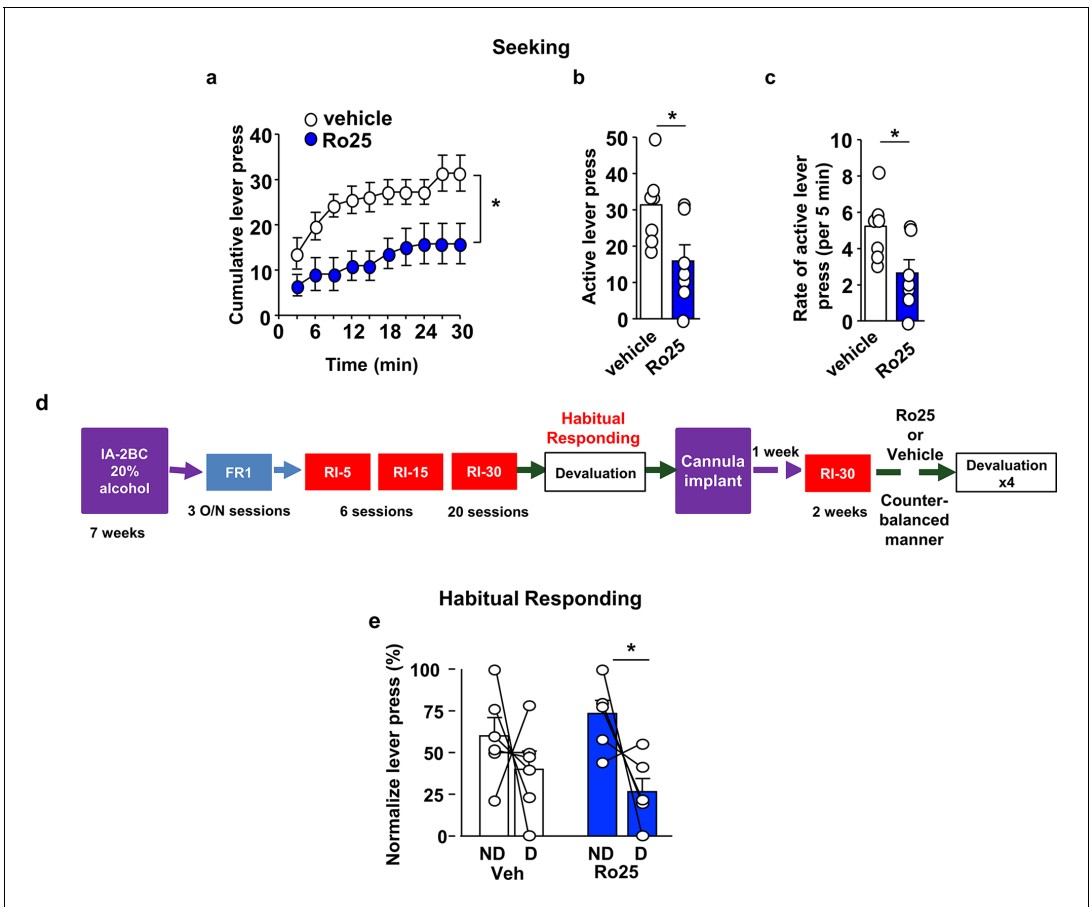

**Figure 5.** GluN2B in the OFC promotes alcohol seeking and habitual alcohol responding. (**a–c**) Intra-OFC administration of Ro25-6981 decreases alcohol seeking. Rats underwent 7 weeks of IA-20%2BC, and were then trained to self-administer 20% alcohol. Vehicle (white) or Ro25-6981 (Ro25, 10 µg/µl, blue) was infused bilaterally in the OFC 15 min prior to a 30 min extinction session. (**a**) Cumulative lever presses. Two-way RM ANOVA revealed a significant main treatment effect ($F_{1,12}=10.89$, $p<0.05$). Two-tailed paired t-test revealed that the number (**b**) and the rate (per 5 min) (**c**) of lever press were significantly different in the vehicle vs. Ro25-6981 groups (both $t_6 = 2.62$, $p<0.05$). (**d**) Timeline of experiment. Rats underwent 7 weeks of IA20%−2BC, and were then trained to operant self-administer 20% alcohol using a progressive RI reinforcement schedule. Following stable RI-30 lever presses, rats underwent stereotaxic surgery to bilaterally implant guide cannulae in the OFC. One week later, RI-30 training was resumed for 2 weeks, after which rats received microinjections of vehicle (white) or Ro25-6981 (5 µg/µl, blue) in a counter-balanced manner 15 min prior to a 30 min home cage 20%2BC alcohol drinking session, and the number of lever presses were measured during extinction. (**e**) Intra-OFC administration of Ro25-6981 attenuates habitual alcohol seeking. Two way RM ANOVA did not detect a main effect of devaluation ($F_{1,5}=4.162$, $p=0.0969$) or a treatment x devaluation interaction ($F_{1,5}=1.956$, $p=0.2208$). Sidak's multiple comparison test detected a significant difference for Ro25-6981 ($p<0.05$) on D compared to ND days. Individual data points and mean ± SEM are shown, (**a–c**) n = 7, (**e**) n = 6. *$p<0.05$.

The online version of this article includes the following source data and figure supplement(s) for figure 5:

**Source data 1.** Cumulative lever presses at 3 min intervals for vehicle- and Ro25-treated rats during a 30 min extinction session (*Figure 5a*).

**Source data 2.** Total lever presses and lever pressing rate (presses/min) during a 30 min extinction session in vehicle- and Ro25-treated rats (*Figure 5b–c*).

**Source data 3.** Total lever presses during 10 min extinction sessions on non-devalued and devalued days in vehicle- and Ro25-treated, RI-trained rats (*Figure 5d*).

**Figure supplement 1.** Schematic drawing of cannulae placement.

**Figure supplement 2.** Inhibition of GluN2B in the OFC does not alter locomotion.

**Figure supplement 2—source data 1.** Inter-response intervals during a 30 min extinction session in vehicle- and Ro25-treated rats.

**Figure supplement 3.** Inhibition of GluN2B in the OFC does not alter voluntary alcohol intake prior to devaluation.

**Figure supplement 3—source data 1.** Alcohol consumed (g/kg) during 30 min home cage alcohol exposure prior to devaluation extinction tests in vehicle- and Ro25-treated rats.

administration. Specifically, prior to the OSA, rats were subjected to 7 weeks of IA-20%2BC. The IA-

20%2BC paradigm models 'problem drinkers' i.e. human subjects that suffer from AUD phenotypes such as binge drinking, compulsive drinking and craving (*Enoch and Goldman, 2002*). In our case, rats achieve blood alcohol concentrations (BAC) of 80 mg/dl (*Carnicella et al., 2014*) a BAC equivalent to binge drinking humans (*NIAAA, 2004*). In contrast, the majority of other reports described herein involved passive exposure of alcohol vapor, in which BAC is 150–250 mg/dl, and in which physical dependence on alcohol is observed (*Griffin, 2014*). Humans who are dependent on alcohol show profoundly degraded white matter and reduced neuronal density in the OFC (*Pfefferbaum and Sullivan, 2005*; *Miguel-Hidalgo et al., 2006*), possibly an indication of alcohol-induced damage to this structure. Thus, it is likely that cycles of binge drinking do not damage the OFC which may be the case in animal models of alcohol dependence. As only a small percentage of alcohol users exhibit physical dependence (*WHO, 2014*), we believe that our findings have important implications for the understanding of the mechanisms underlying problem drinking and AUD.

Our results suggest that the glutamatergic system is recruited during alcohol withdrawal leading to mTORC1 activation in the OFC. One major glutamatergic input in the OFC is the thalamus (*Fresno et al., 2019*). It would be therefore of interest to investigate whether thalamic nuclei inputs are recruited by alcohol to activate mTORC1 in the OFC.

We show here that GluN2B activation in the OFC may be sufficient to influence the formation and/or maintenance of alcohol seeking and habit. In line with the possibility that GluN2B activation contributes to the development of habit, DePoy et al. showed that habitual behavior, developed upon subchronic cocaine treatment during adolescence (*DePoy et al., 2017*), is inhibited by the treatment of animals with the GluN2B inhibitor, ifenprodil (*DePoy et al., 2017*). DePoy et al. further showed that cocaine exposure during adolescence produces structural changes in the OFC which were attenuated by ifenprodil (*DePoy et al., 2017*). While our data suggest that GluN2B in the OFC participates in behavioral inflexibility, Brigman et al. reported that GluN2B in the OFC plays a role in choice learning (*Brigman et al., 2013*). It is plausible that alcohol-dependent neuroadaptations tilt the function of GluN2B from choice learning, for example a goal-directed behavior, to habitual responding.

What could be the mechanism by which mTORC1 contributes to alcohol seeking and habit? Activation of mTORC1 triggers the translation of a subset of mRNAs to protein in the cell body and in dendrites (*Saxton and Sabatini, 2017*; *Yoon et al., 2016*). We previously showed that long-term excessive alcohol intake and reinstatement of alcohol place preference initiate the translation of several synaptic proteins in the NAc (*Ben Hamida et al., 2019*; *Liu et al., 2017*; *Laguesse et al., 2017b*). We also found that reconsolidation of alcohol reward memories increases the immunoreactivity of mTORC1 targets in the OFC (*Barak et al., 2013*). Thus, further studies are required to identified transcripts which are translated in response to alcohol exposure and determine their role in alcohol seeking and habit.

## Materials and methods

### Reagents

Anti-phospho-S6 (S235/236, 1:500), anti-S6 (1:1000), anti-phospho-4E-BP (T37/46, 1:500), anti-4E-BP (1:1000), anti-phospho-CaMKII (T286, 1:1000), anti-phospho-S6K (T389, 1:500), anti-S6K (1:500), anti phospho-GluN2B (Y1472, 1:500) and anti GluN2B (1:1000) antibodies were purchased from Cell Signaling Technology (Danvers, MA). Anti-CaMKII (1:500) antibodies were purchased from Santa Cruz Biotechnology (Santa Cruz, CA). Anti-Actin (1:10,000) antibodies, phosphatase Inhibitor Cocktails 2 and 3, Dimethyl sulfoxide (DMSO), D-sucrose, tetrodotoxin (TTX) and NMDA were purchased from Sigma Aldrich (St. Louis, MO). Nitrocellulose membranes were purchased from EMD Millipore (Billerica, MA, USA). Enhanced Chemiluminescence (ECL) was purchased from GE Healthcare (Pittsburg, PA, USA). Donkey anti-rabbit horseradish peroxidase (HRP), and donkey anti-mouse horseradish peroxidase (HRP) were purchased from Jackson ImmunoResearch (West Grove, PA). EDTA-free complete mini Protease Inhibitor Cocktails were purchased from Roche (Indianapolis, IN). NuPAGE Bis-Tris precast gels and Phosphate buffered saline (PBS) were purchased from Life Technologies (Grand Island, NY). Bicinchoninic Acid (BCA) protein assay kit was obtained from Thermo Scientific (Rockford, IL), and ProSignal Blotting Film was purchased from Genesee Scientific (El Cajon, CA). Ethyl

alcohol (190 proof) was purchased from VWR (Radnor, PA). Rapamycin was purchased from LC laboratories (Woburn, MA) and Ro25-6981 was purchased from Tocris Bioscience (Bristol, UK).

## Subjects

Male Long Evans rats (Harlan, Indianapolis, IN) were one (ex vivo experiment) or two months (in vivo experiments) old at their arrival. Animals were single-housed in a temperature- and humidity-controlled colony room (22 ± 2°C, relative humidity: 50–60%) under a normal 12 hr light/dark cycle (lights on at 7:00AM) with food and water available ad libitum. Rats were given a week of habituation to the housing conditions before the beginning of the experiments. All animal procedures were approved by the University of California San Francisco Institutional Animal Care and Use Committee (IACUC) (protocol number number AN179720-01B), and were conducted in agreement with the Association for Assessment and Accreditation of Laboratory Animal Care (AAALAC).

## Preparation of solutions

Alcohol was diluted to 20% (v/v) and sucrose to 1–8% (w/v) in tap water. Rapamycin was dissolved in PBS. Ro25-6981 and NMDA were dissolved in 0.1% DMSO in PBS.

## Intermittent access to 20% alcohol using two bottle choice and drug infusion

Rats underwent intermittent access to 20% alcohol in a 2-bottle choice (IA-20%2BC) paradigm for 7 weeks as described in *Laguesse et al. (2017b)*. Specifically, rats were given 24 hr of concurrent access to one bottle of 20% alcohol (v/v) in tap water and one bottle of water. Control rats had access to water only. Drinking sessions started at 12:00pm on Monday, Wednesday and Friday, with 24- or 48 hr (weekend) of alcohol-deprivation periods in which rats consumed only water. The placement (left or right) of the water or alcohol solution was alternated between each session to control for side preference. Water and alcohol bottles were weighed at the beginning and at the end of each alcohol drinking session. Rats were weighed once a week. Rats drinking more than 3.5 g/kg/24 hr on the last week of IA-20%2BC were selected for the study.

## Stereotaxic surgery

Following a stable baseline of alcohol drinking, rats underwent stereotaxic surgery. Specifically, rats were continuously anesthetized using isoflurane (Baxter) and bilateral guide cannulae (Plastic One) were implanted in the ventrolateral OFC (anteroposterior +3.5 mm, mediolateral ±2 mm and dorsoventral +3.9 mm; all coordinates are from bregma), secured with screws (Plastic One) and dental cement (Ortho-Jet, Lang Dental).

## Alcohol seeking and drug infusion

First, rats underwent IA-20%2BC paradigm for 7 weeks, as described above. Rats drinking more than 3.5 g/kg/24 hr on the last week of IA-20%2BC were selected for alcohol self-administration training in an operant self-administration (OSA) paradigm (*Jeanblanc et al., 2009*) with modification adapted from *Gremel et al. (2016)*; *Corbit et al. (2012)*; *Renteria et al. (2018)*. Training was conducted during the light cycle in operant chambers (Med-Associates; Georgia, VT) using a fixed ratio 3 (FR3) (i.e. three active lever presses result in the delivery of one alcohol reward) during daily a daily 30 min session (Mon-Fri) as previously described (*Jeanblanc et al., 2009*). An inactive lever was also extended during the operant sessions but had no programmed consequence. Following stable baseline of alcohol responding, rats underwent stereotaxic surgery for cannula placement in the OFC, as described above. After a one-week recovery period, operant self-administration procedure was resumed and handling for drug microinfusions began. Rats received infusion of either vehicle or rapamycin (50 ng/μl/side) 3 hr or R025-6981 (10 μg/μl/side) 15 min before a single 30 min extinction session. Drugs were injected with bilateral infusion needles (Plastics One) that projected 0.5 mm past the end of the guide cannula, at a rate of 0.5 μl/minute. Drug testing was performed using a 'within-subject' design, in which rats received both treatments in counterbalanced order.

Upon establishing a baseline of IA-20%2BC, a separate cohort of animals received an infusion of R025-6981 (10 μg/μl/side) or vehicle 15 min prior to the end of the last withdrawal session. The OFC

was harvested at the end of the 24 hr withdrawal session and were processed for western blot analysis.

## Habitual and goal-directed alcohol seeking and drug infusion

After 7 weeks of IA-20%2BC, high-drinking rats (alcohol intake >3.5 g/kg/24 hr) were trained to self-administer 20% alcohol in an OSA paradigm as described above. Self-administration training was initiated under fixed ration 1 (FR1) for three consecutive overnight sessions. Afterwards, the sessions lasted for an hour and took place during the light cycle. Only one lever was available during the operant sessions. Rats were pseudo-randomly assigned to two groups and were subjected to a random interval (RI), or a random ratio (RR) schedule of reinforcement which biases responding towards habitual or goal directed actions, respectively (*Gremel and Costa, 2013*). Only rats that self-administered alcohol more than 0.3 g/kg/hour were included in the study.

RI training: During RI training, each reward was delivered following one lever press occurring after variable reward delivery time intervals (*Gremel and Costa, 2013*). Three sessions were under RI-5 (i.e. inter-reward time was 5 s on average, ranging from 1 to 10 s), followed by three RI-15 sessions (i.e. inter-reward time was 15 s on average ranging from 10 to 30 s). Reinforcement was then shifted to RI-30 (i.e. inter-reward time was 30 s on average ranging from 10 to 50 s) for at least 20 sessions (Timeline, *Figure 2a*).

RR training: During RR training, reward was delivered following a variable number of lever responses (*Gremel and Costa, 2013*). Six sessions under RR-2 (one reward delivery following on average two lever presses with number of presses ranging from 1 to 3) followed by RR-3 (one reward delivery following on average three lever presses with number of presses ranging from 1 to 5) for at least 20 sessions (Timeline, *Figure 2a*).

Following stable RI-30 or RR-3 responding, rats underwent alcohol devaluation testing as described in *Corbit et al. (2012)* with modifications. Rats were given home-cage 2BC access to 1% sucrose on non-devalued (ND) days or 20% alcohol on devalued (D) days for 30 min (*Figure 2b*). On each day, immediately after home-cage pre-feeding, rats underwent a 10 min extinction test during which unrewarded lever presses were recorded. A 'within-subject' design was used in which ND or D sessions were in a counterbalanced order, with two standard retraining sessions between the tests. To minimize the effects of individual variance in total lever pressing, we normalized ND and D lever presses (*Baltz et al., 2018*; *Gremel and Costa, 2013*; *Gremel et al., 2016*). Normalized lever pressing was calculated as ND or D lever presses/total number of lever presses, respectively. Normalizing devaluation lever presses produces a distribution of data, where 50% is equivalent to equal lever pressing between ND and D (*Gremel and Costa, 2013*). At the end of the training period and upon acquiring the behavior (RR, goal directed behavior and RI, habitual behavior), rats underwent stereotaxic surgery to bilaterally implant a guide cannula in the ventrolateral OFC, as described above. One week later, RI-30 or RR-3 training was resumed for 2 weeks after which rats received microinjections of either vehicle or rapamycin (50 ng/µl/side) 3 hr or R025-6981 (5 µg/µl/side) 15 min prior an alcohol devaluation testing.

## Habitual sucrose seeking

Habitual sucrose training followed an approach similar to that described above for habitual alcohol seeking. After 7 weeks of IA-20%2BC, high-drinking rats (alcohol intake >3.5 g/kg/24 hr) were trained to self-administer 8% sucrose in an OSA paradigm. RI-30 training was similar to the procedure described for alcohol except that 8% sucrose was initially used and progressively reduced to 1% across training sessions. Only ten RI30 sessions (2 weeks) were conducted prior to the initial outcome devaluation test. Rats were then implanted with a guide cannula in the OFC as described above. After a week recovery period, sucrose self-administration was resumed for two weeks, after which rats received microinjections of vehicle or drug in a counter-balanced manner prior to the devaluation test. Sensitivity to changes in sucrose value was tested using a sucrose devaluation procedure which was similar to alcohol devaluation, except that rats had 2BC access to 20% alcohol on ND day and 1% sucrose on D day (Timeline, *Figure 3a–b*).

## Histology

For verification of cannula placement, brains were post-fixed in 4% PFA for one week, then rapidly frozen and sectioned into 50 µm coronal slices using a Leica CM3050 cryostat (Leica Biosystems). Slices were stained using cresyl violet and examined for cannula placement using a bright-field microscope. Rats with correct guide cannula placement were included in the study.

## Ex vivo NMDA treatment

Naïve rats were euthanized by isoflurane and decapitated. Brains were removed and sectioned into 300 µm-thick slices at 4°C in a solution of aCSF containing (in mM) 200 sucrose, 1.9 KCl, 1.4 $NaH_2PO_4$, 0.5 $CaCl_2$, 6 $MgCl_2$, 10 glucose, 25 $NaHCO_3$, four ascorbic acid; 310–330 mOsm. Immediately after sectioning, OFC-containing slices were recovered in aCSF (in mM: 125 NaCl, 2.5 KCl, 1.4 $NaH_2PO_4$, 25 $NaHCO_3$, 2 $CaCl_2$, 1.3 $MgCl_2$, 25 glucose, 0.4 ascorbic acid) heated at 32°C for 30 min and then moved to an aCSF solution containing 1 µM TTX at room temperature for 30 min (*Sutton and Chandler, 2002*). The OFC was dissected and incubated for 30 min in aCSF containing 1 µM TTX at room temperature. OFC sections were then treated for 3 min with vehicle (0.1% DMSO) or NMDA (25 µM) in TTX-containing aCSF , and western blot analysis was conducted as described below.

## Collection of OFC for western blot

Animals were euthanized and brains rapidly dissected on ice using a 1 mm brain block. One mm thick coronal sections located between +4.20 mm and +3.20 mm anterior to bregma were removed and the OFC was dissected on an anodized aluminum block on ice, using the Paxinos and Watson stereotaxic atlas 4[th] edition as a reference. Tissue was collected into 1.5 ml Eppendorf tubes and immediately homogenized in 300 µl RadioImmuno Precipitation Assay (RIPA) buffer (in mM: 50 Tris-HCl, pH 7.6, 150 NaCl, 2 EDTA, and 1% NP-40, 0.1% SDS and 0.5% sodium deoxycholate, protease and phosphatase inhibitor cocktails).

## Western blot analysis

Tissue were homogenized in ice-cold RIPA buffer , using a sonic dismembrator. Protein content was determined using BCA kit. Tissue homogenates (30 µg per sample) were resolved on NuPAGE 10% Bis-Tris gels at 100 V for 2 hr and transferred onto nitrocellulose membranes at 30V for 2 hr. Blots were blocked with 5% milk-phosphate-buffered saline with 0.1% tween-20 at room temperature and then probed with primary antibodies overnight at 4°C. Membranes were washed and probed with HRP-conjugated secondary antibodies for 2 hr at room temperature, and bands were visualized using ECL. Band intensities were quantified by ImageJ (National Institutes of Health, MD, USA).

## Data analysis

All data are expressed as the mean ± SEM. GraphPad Prism 7.0 (GraphPad Software, Inc, La Jolla, CA, USA) was used to plot and analyze the data. D'Agostino–Pearson normality test and *F*-test/Levene test were used to verify the normal distribution of variables and the homogeneity of variance, respectively. The sample sizes were based on previous publications (*Neasta et al., 2010*; *Barak et al., 2013*). Data between two groups were compared using two-tailed paired or unpaired *t*-test. Data from multiple groups against one group were compared using a two-way repeated-measures analysis of variance (RM-ANOVA) followed when appropriate by a Tukey multiple comparisons test or a Sidak's multiple comparisons test. Statistical significance was set at $p < 0.05$.

## Acknowledgements

This research was supported by the National Institute of Alcohol Abuse and Alcoholism, P50 AA017072 (DR) and R01 AA027474 (DR).

## Additional information

### Funding

| Funder | Grant reference number | Author |
|---|---|---|
| National Institute on Alcohol Abuse and Alcoholism | P50 AA017072 | Dorit Ron |
| National Institute on Alcohol Abuse and Alcoholism | R01 AA027474 | Dorit Ron |

The funders had no role in study design, data collection and interpretation, or the decision to submit the work for publication.

### Author contributions

Nadege Morisot, Conceptualization, Data curation, Formal analysis, Investigation, Methodology; Khanhky Phamluong, Data curation, Formal analysis, Investigation; Yann Ehinger, Data curation, Formal analysis, Validation, Methodology; Anthony L Berger, Data curation, Investigation; Jeffrey J Moffat, Formal analysis; Dorit Ron, Conceptualization, Supervision, Funding acquisition, Project administration

### Author ORCIDs

Yann Ehinger (ID) https://orcid.org/0000-0001-9314-6575
Dorit Ron (ID) https://orcid.org/0000-0001-5161-967X

### Ethics

Animal experimentation: All animal procedures were approved by the University of California San Francisco Institutional Animal Care and Use Committee (IACUC) and were conducted in agreement with the Association for Assessment and Accreditation of Laboratory Animal Care (AAALAC).(protocol number number AN179720-01B).

### Decision letter and Author response

Decision letter https://doi.org/10.7554/eLife.51333.sa1
Author response https://doi.org/10.7554/eLife.51333.sa2

## Additional files

### Supplementary files

• Transparent reporting form

### Data availability

Source data files have been provided for Figures 1-5.

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
