## [Decision Letter]

**Acceptance summary:**

This study defines a novel role for mechanistic target rapamycin complex 1 (mTORC1) action, potentially through GluN2B activation, in the orbitofrontal cortex (OFC) in alcohol seeking and habitual intake. The study demonstrates that inhibition of mTORC1 activity attenuates alcohol seeking and restores sensitivity to outcome devaluation in rats that habitually seek alcohol. This effect was specific to alcohol as mTorc1 inhibition did not alter habitual responding to a natural reward. Overall, the study provides new insight into a mechanism, via GluN2B and mTORC1, in mediating a shift from goal-directed to habitual alcohol seeking.

**Decision letter after peer review:**

Thank you for submitting your article "mTORC1 in the orbitofrontal cortex promotes habitual alcohol seeking" for consideration by *eLife*. Your article has been reviewed by Kate Wassum as the Senior Editor, a Reviewing Editor, and three reviewers. The following individuals involved in review of your submission have agreed to reveal their identity: Michelle Mazei-Robison (Reviewer #3).

The reviewers have discussed the reviews with one another and the Reviewing Editor has drafted this decision to help you prepare a revised submission.

Essential revisions:

While the reviewers find the study to be of interest, they raise concerns that must be adequately addressed before the paper can be accepted. The reviewers agree that these revisions should not require additional experiments.

1) Please provide a compelling rationale for the exclusion criteria and provide details about numbers excluded further clarity on the exclusion criterion (did not acquire the behavior is vague) and why the exclusions do not create bias. This is in reference to the following statement: "did not acquire the behavior and were pressing less than 5 lever presses per session on ND day" (subsection “Habitual and goal-directed alcohol seeking and drug infusion”).

2) Please address fundamental issues with the below pattern of results, which appear to be empirically and conceptually discordant. This will aid the reader in understanding these discrepancies and clarify the overall significance of the current findings.

a) Please address how a manipulation disrupts alcohol seeking under normal conditions (without any satiety treatment, Figure 1D,E,F and Figure 4A,B,C) but not during ND tests (when rats are sated on irrelevant outcome, Figure 2E,G and Figure 5E). Could important procedural differences help to explain these differences?

b) Please address how disrupting habitual control can lead to a disruption of alcohol seeking, if the current and published (e.g., Yin et al., 2004; Corbit et al., 2014) data indicates that this does not lead to a response decrement, apparently due to compensation by the goal-directed system. While not all findings must fit tightly these issues are fundamental to judging the significance of the findings and their value to the field.

3) Please ensure that similar and appropriate details about instrumental training procedures are provided for all experiments, including differences; e.g., subsection “Alcohol seeking and drug infusion”. For instance, one difference is that rats are sated on sucrose prior to ND tests, while another difference is test duration.

4) Please provide a rationale for why phosphorylated GluN2B was not measured? The data in Figure 4C-J nicely links GluN2B function to alcohol-induced mTORC1 and the authors address alcohol-induced changes in GluN2B phosphorylation in other brain regions in the discussion. However, the activation of GluN2B is not specifically addressed in the study.

5) In Figure 2E, post-hoc comparisons are not necessary because a main effect between two conditions was found (not an interaction). Thus, the main effect can be indicated in the figure to emphasize that the groups don't differ in their preference for the valued outcome, but the direct comparisons should be eliminated.

[Editors' note: further revisions were requested prior to acceptance, as described below.]

Thank you for resubmitting your work entitled "mTORC1 in the orbitofrontal cortex promotes habitual alcohol seeking" for further consideration by *eLife*. Your revised article has been evaluated by Kate Wassum (Senior Editor) and a Reviewing Editor.

The manuscript has been improved but there are some remaining issues that need to be addressed before acceptance, as outlined below:

1) The reviewers felt that the discussion was unfocused and speculative. While they do understand that this may reflect the goal to address the previous concerns, a more focused discussion would improve the manuscript. Below are the concerns that the reviewers felt should be addressed:

a) Discussion section, "reward anticipation" is invoked. This is problematic for the habit argument because habits are by nature uncoupled from reward value and an anticipation of the outcomes of one's behaviors. Thus, reward anticipation would not be a factor driving habitual behavior. If the authors want to make this argument, this should be clearly dissociated from the habit argument. Alternatively, rather than referring to action-outcome based reward anticipation, the authors may be referring to context-alcohol learning, which is likely a major source of motivation for habitual alcohol seeking. It is known that natural reward-paired cues preferentially motivate habitual reward-seeking actions (Holland, 2004; Wilgen et al., 2012) in a manner that does not depend on reward value (Rescorla, 1994; Holland, 2004).

b) Discussion section, the authors argue that mTORC1 could be strengthening OFC projections to the DLS. These projections do not exist – the OFC projects to large swaths of the striatum, but not the habit-associated DLS, referring to the dorsal-most region of the lateral striatum.

c) Discussion section remarks, "it would be of interest to determine the normal role of mTORC1, and whether mTORC1 suppresses other decision-making tasks." This line was confusing given the author's investigations with sucrose reinforcers assessed this prospect.

d) Discussion section concludes with the phrase "with less involvement coming from the lOFC." The cited references (Bradfield, Hart and Balleine, 2018; Gourley et al., 2016) do not investigate the lOFC, and thus, do not provide evidence for this notion. This section as a whole appears unfocused. It is suggested that this section be significantly edited down to simply make the point that the lOFC and mOFC have different functions in the context of goal-directed behavior. Currently, the arguments regarding the balance between the two are difficult to follow and very speculative, given the sole focus on lOFC here.

e) The authors' responses to comments 2 and 3 touch on important issues that help address confusion in understanding some apparent discrepancies in the data. However, the pieces are not entirely put together. The authors suggest that rapamycin and Ro25 treatments disrupt alcohol seeking late in nonreinforced tests, which explains why no deficits are found in short ND tests (also nonreinforced). However, the significance of this is not addressed. The reviewers suggest explicitly raising the possibility that habit processes may preferentially contribute to persistence of alcohol seeking in extinction. While persistence in extinction is not a critical test of habit behavior, it is certainly consistent with this account. Providing such a discussion would help link these findings to the devaluation results, which would benefit many readers.

2) There were concerns with the exclusion of subjects based on arbitrarily low performance of the nondevalued action (only) at test because this, by design, creates a bias, increasing the likelihood of detecting a difference between devalued vs. nondevalued actions. The rationale was that low nondevalued action performance at test meant that animals had "generally not escalated their lever-pressing during training". Based on this it is suggested that the authors exclude based on low response rate during training, which doesn't create a systematic bias and is in line with accepted practice. Alternatively, it suggested that the authors include the two excluded subjects.

---

## [Author Response]

Essential revisions:While the reviewers find the study to be of interest, they raise concerns that must be adequately addressed before the paper can be accepted. The reviewers agree that these revisions should not require additional experiments.1) Please provide a compelling rationale for the exclusion criteria and provide details about numbers excluded further clarity on the exclusion criterion (did not acquire the behavior is vague) and why the exclusions do not create bias. This is in reference to the following statement: "did not acquire the behavior and were pressing less than 5 lever presses per session on ND day" (subsection “Habitual and goal-directed alcohol seeking and drug infusion”).

This is an important point, which we have addressed by indicating the number of excluded animals for each experiment in the figure legends and clarifying the reasoning for the exclusion criteria in the Materials and methods section. In essence, this study is focused on alcohol seeking behaviors, and we have attempted to exclude animals from the study who do not reliably seek alcohol. We set the cutoff at 5 lever presses, because animals who press at this rate have generally not escalated their lever-pressing during training, which could mean that they have not properly learned the task or may not find alcohol sufficiently rewarding to elicit seeking behavior. Including these animals can also skew the normalized lever pressing data in either direction due to very slight changes in pressing, much smaller than the standard deviation. Only two rats were excluded from the study.

2) Please address fundamental issues with the below pattern of results, which appear to be empirically and conceptually discordant. This will aid the reader in understanding these discrepancies and clarify the overall significance of the current findings.a) Please address how a manipulation disrupts alcohol seeking under normal conditions (without any satiety treatment, Figure 1D,E,F and Figure 4A,B,C) but not during ND tests (when rats are sated on irrelevant outcome, Figure 2E,G and Figure 5E). Could important procedural differences help to explain these differences?

We thank the reviewers for raising this question. It is important to note that extinction does disrupt alcohol seeking in general, with or without satiety treatment. One difference between the extinction tests is that tests depicted in Figure 1 and Figure 5A last 30 minutes, while the extinction tests in Figure 2 and Figure 5E (devaluation tests) last only 10 minutes. In order to minimize any general effects of satiety treatment, none of the rats in any of the experiments were food or water restricted at any time. Although it is possible that the home cage satiety treatment on an irrelevant outcome (ND tests) could impact seeking behavior, we did not compare alcohol seeking in these animals with an additional group that underwent no satiety treatment, as this experiment is beyond the scope of the questions addressed in this manuscript. We believe that even if satiety treatment on an irrelevant outcome does impact seeking in either direction, this would not change the main conclusions drawn from the data presented here.

b) Please address how disrupting habitual control can lead to a disruption of alcohol seeking, if the current and published (e.g., Yin et al., 2004; Corbit et al., 2014) data indicates that this does not lead to a response decrement, apparently due to compensation by the goal-directed system. While not all findings must fit tightly these issues are fundamental to judging the significance of the findings and their value to the field.

This comment raises a salient point on the impact of shifting action control from a more habitual to goal-directed strategy in general. As far as translational goals are concerned, we are not preoccupied with necessarily reducing alcohol consumption in all cases, but rather with decreasing habitual, excessive, and compulsive drinking. While the overall treatment goal with other drugs of abuse may be complete abstinence, alcohol is unique in that moderate alcohol consumption under acceptable circumstances meets societal standards in many regions of the world. For this reason, more research is needed into the respective roles of habitual and goal-directed alcohol seeking in human cohorts. It is likely that there is at least some compensation from the goal-directed system when you disrupt habitual control. Indeed, we show in Figure 1 that inhibition of mTOR in the OFC does not alter alcohol consumption or self-administration except in extinction sessions, which would indicate that these animals still drink at the same level, but their action selection strategy has shifted.

3) Please ensure that similar and appropriate details about instrumental training procedures are provided for all experiments, including differences; e.g., subsection “Alcohol seeking and drug infusion”. For instance, one difference is that rats are sated on sucrose prior to ND tests, while another difference is test duration.

We have updated the Materials and methods section of the paper to minimize confusion between the different devaluation protocols for sucrose and alcohol.

4) Please provide a rationale for why phosphorylated GluN2B was not measured? The data in Figure 4C-J nicely links GluN2B function to alcohol-induced mTORC1 and the authors address alcohol-induced changes in GluN2B phosphorylation in other brain regions in the discussion. However, the activation of GluN2B is not specifically addressed in the study.

We thank the reviewer for the suggestion. Not measuring GluN2B phosphorylation was an oversight on our behalf especially since it is a true indication of GluN2B activation. We added new data to the revision showing that alcohol significantly increases GluN2B phosphorylation with no accompanying change in total GluN2B protein level. These results are now presented in Figure 4.

5) In Figure 2E, post-hoc comparisons are not necessary because a main effect between two conditions was found (not an interaction). Thus, the main effect can be indicated in the figure to emphasize that the groups don't differ in their preference for the valued outcome, but the direct comparisons should be eliminated.

Per the reviewer’s request, we revised the statistics and figure legend for Figure 2E (now Figure 2D).

[Editors' note: further revisions were requested prior to acceptance, as described below.]

The manuscript has been improved but there are some remaining issues that need to be addressed before acceptance, as outlined below:1) The reviewers felt that the discussion was unfocused and speculative. While they do understand that this may reflect the goal to address the previous concerns, a more focused discussion would improve the manuscript. Below are the concerns that the reviewers felt should be addressed:a) Discussion section, "reward anticipation" is invoked. This is problematic for the habit argument because habits are by nature uncoupled from reward value and an anticipation of the outcomes of one's behaviors. Thus, reward anticipation would not be a factor driving habitual behavior. If the authors want to make this argument, this should be clearly dissociated from the habit argument. Alternatively, rather than referring to action-outcome based reward anticipation, the authors may be referring to context-alcohol learning, which is likely a major source of motivation for habitual alcohol seeking. It is known that natural reward-paired cues preferentially motivate habitual reward-seeking actions (Holland, 2004; Wilgen et al., 2012) in a manner that does not depend on reward value (Rescorla, 1994; Holland, 2004).

We appreciate the reviewers’ point and the validity of their argument. We modified the term “action-outcome based reward anticipation” to “context-alcohol learning”.

b) Discussion section, the authors argue that mTORC1 could be strengthening OFC projections to the DLS. These projections do not exist – the OFC projects to large swaths of the striatum, but not the habit-associated DLS, referring to the dorsal-most region of the lateral striatum.

We have removed this part of the Discussion section.

c) Discussion section remarks, "it would be of interest to determine the normal role of mTORC1, and whether mTORC1 suppresses other decision-making tasks." This line was confusing given the author's investigations with sucrose reinforcers assessed this prospect.

We apologize for the confusion. By normal we are referring to known OFC-dependent behaviors such as decision making (Wallis, 2007; Meyer and Bucci, 2016; Baltz et al., 2018), stimulus-outcome association (Ostlund and Balleine, 2007; Gourley et al., 2013; Gremel and Costa, 2013) and updating the value of predicted outcomes (Fiuzat et al., 2017; Baltz et al., 2018). The word “normal” has been deleted from the text.

d) Discussion section concludes with the phrase "with less involvement coming from the lOFC." The cited references (Bradfield, Hart and Balleine, 2018; Gourley et al., 2016) do not investigate the lOFC, and thus, do not provide evidence for this notion. This section as a whole appears unfocused. It is suggested that this section be significantly edited down to simply make the point that the lOFC and mOFC have different functions in the context of goal-directed behavior. Currently, the arguments regarding the balance between the two are difficult to follow and very speculative, given the sole focus on lOFC here.

This part of the Discussion section was edited according to the reviewers’ suggestion.

e) The authors' responses to comments 2 and 3 touch on important issues that help address confusion in understanding some apparent discrepancies in the data. However, the pieces are not entirely put together. The authors suggest that rapamycin and Ro25 treatments disrupt alcohol seeking late in nonreinforced tests, which explains why no deficits are found in short ND tests (also nonreinforced). However, the significance of this is not addressed. The reviewers suggest explicitly raising the possibility that habit processes may preferentially contribute to persistence of alcohol seeking in extinction. While persistence in extinction is not a critical test of habit behavior, it is certainly consistent with this account. Providing such a discussion would help link these findings to the devaluation results, which would benefit many readers.

This comment raises an important point. While the reviewers are correct in noting that persistent alcohol seeking during extinction may be consistent with habitual behavior, we do not have sufficient data to support this claim, as the animals in Figure 1 and Figure 5A-C did not undergo RI training or even extended FR training, which would condition untreated rats toward more habitual seeking. The differences in seeking behavior between rapamycin- or Ro25-treated animals in the 30-minute nonreinforced tests are likely to be due to, at least in part, to changes in action-selection strategy, but speculation on enhancing already goal-directed behavior might generate uncertainty in readers. We addressed this concern in the Discussion section by highlighting the possibility that a shift in action-selection strategy may already be present in the 30-minute nonreinforced sessions, but we need to use RI training and devaluation to confirm this.

2) There were concerns with the exclusion of subjects based on arbitrarily low performance of the nondevalued action (only) at test because this, by design, creates a bias, increasing the likelihood of detecting a difference between devalued vs. nondevalued actions. The rationale was that low nondevalued action performance at test meant that animals had "generally not escalated their lever-pressing during training". Based on this it is suggested that the authors exclude based on low response rate during training, which doesn't create a systematic bias and is in line with accepted practice. Alternatively, it suggested that the authors include the two excluded subjects.

Per the reviewers’ request, we now include in the analysis the two rats which are pressing below threshold. Inclusion of the subjects did not change the interpretations and/or conclusions, the text and figures have been updated accordingly.